# Socio-Ecological Model of Correlates of Double Burden of Malnutrition in Developing Countries: A Narrative Review

**DOI:** 10.3390/ijerph16193730

**Published:** 2019-10-03

**Authors:** Trias Mahmudiono, Calista Segalita, Richard R. Rosenkranz

**Affiliations:** 1Department of Nutrition, Faculty of Public Health, Universitas Airlangga, Jl. Mulyorejo Kampus C, Surabaya 60115, Indonesia; calsglt@gmail.com; 2Department of Food, Nutrition, Dietetics & Health, Kansas State University, Manhattan, KS 66506, USA; ricardo@ksu.edu

**Keywords:** double burden of malnutrition, socio-ecological model, nutrition transition, stunting, underweight

## Abstract

(1) Background: The double burden of malnutrition (DBM) is a complex problem involving the coexistence of under- and over-nutrition within the same individual, household or population. This review aimed to discuss the correlates of the double burden of malnutrition through the socio-ecological model (SEM); (2) Methods: The PubMed database was systematically searched for peer-reviewed articles related to the double burden of malnutrition. Information on correlates of the double burden of malnutrition was extracted for analysis and discussion in alignment with the levels of the socio-ecological model.; (3) Results and Discussion: The correlates of the double burden of malnutrition identified from previous literature were: Race/genetics; maternal short stature; breastfeeding status; low maternal education; family size; household food security; household dietary diversity; and rural and urban settings. In the absence of evidence linking factors in a certain level of the SEM and the double burden of malnutrition, we employed correlates of overweight status and obesity to complete this narrative. Potential intervention strategies were proposed in alignment with the targets and settings identified, based on the socio-ecological approach; (4) Conclusions: The double burden of malnutrition is a public health phenomenon associated with a variety of socio-ecological determinants. An integrated approach is needed to address the root causes of malnutrition in all its forms, and at all life stages.

## 1. Introduction

One of the primary public health problems of the 21st century is the obesity epidemic that affects more than half a billion people worldwide [1]. The magnitude of the obesity problem is overwhelming in high-income and developed countries. It is estimated that by 2008, 1.46 billion adults were overweight (body-mass index ((BMI) > 25 kg/m^2^), and that 205 million men and 297 million women among them were obese (BMI > 30 kg/m^2^) [2]. In 2016, more than 1.9 billion adults (18 years and older), were overweight and over 65 million were obese, the equivalent of 39% and 13%, respectively [3]. Among developed countries, the USA had the highest prevalence of obesity [4]. The trend in the prevalence of obesity among youth and adults in the U.S. between 2003 and 2004 and 2011 and 2012 remains high (above 30%) and has not changed significantly (point change = 2.8%; 95% confidence interval (CI) = −0.8% to 6.4%) [5] despite a substantive response by the Centers for Disease Control and Prevention (CDC) since 1999 [6]. Moreover, worldwide obesity prevalence for children and adolescents (aged 5–19 years) has risen dramatically from just 4% in 1975 to over 18% in 2016 [3].

Obesity does not only affect developed countries; developing countries also suffer from obesity at an increasing pace [1]. Data from the World Health Organization (WHO) from 1981 to 2008 showed that the prevalence of overweight status and obesity in Africa and Southeast Asia were relatively lower than more developed countries, at 6.5% in countries with lower-middle income and 24.5% in upper-middle income countries [1], but it is expected that these countries will soon face the levels of overweight people that are prevalent in developed countries such as the USA [7]. On the other hand, the prevalence of under-nutrition is far from insignificant. Even though the United Nation’s efforts to combat malnutrition through the Millennium Development Goals (MDGs) are progressing in the right direction [8], almost half of all deaths among children under the age of 5 years are still attributable to undernutrition [9]. The United Nations Children’s Fund (UNICEF) reported that in 2011 more than one-quarter of world’s children under 5 years of age were stunted and 16% were underweight [10]. Recent reports have indicated that stunting has declined from one in three to just one in four between 2000 and 2018 [11]. Although the proportion is decreasing, it is still considered high, due to the public health burden.

In developing nations, nutrient deficiencies manifesting as under-nutrition (underweight, wasting and stunting) persist, while the problem of overweight people and obesity has increased rapidly. Reporting on studies in six developing countries, the Food and Agriculture Organization (FAO) of the United Nations [12] referred to this phenomenon as the double burden of malnutrition (DBM), where under- and over-nutrition occur simultaneously among distinct population groups in developing countries.

Existing studies have defined the double burden of malnutrition at the individual, household, and population or country level [13]. At the individual level, the problems of under- and over-nutrition may coexist in the same individual. An example would be an individual who is obese and suffering from iron deficiency anemia simultaneously [14]. One study in Mexico defined the double burden of malnutrition at the individual level as the concurrence of child stunting and overweight/obesity occurring with anemia in children and in women [15]. At the household level, the problem occurs when one member is undernourished while another is overweight. Some studies have highlighted the phenomena of a stunted child and overweight or obese mother in the same household. Unfortunately, there is a lack of consistency and agreement on terminology. Several names have been used to represent this phenomenon including SCOWT (stunted child and overweight mother pairs) [16], SCOM (stunted child and overweight/obese mother) [17], DBM [18] and MCDB (maternal child double burden) [19]. A recent study in Ecuador reported an investigation of the double burden of malnutrition phenomenon across a decade; it was found that there was positive association between overweight/obese male and child stunting within the community level, but not at the household level [20]. Furthermore, this study suggested that interventions will be more effective if targeted at the community, instead of the household level alone.

Most studies in the peer-reviewed literature describe the double burden of malnutrition at the population or country level, where a portion of the population suffers from undernutrition while another part of the population faces the problem of over-nutrition. At the national level, the double burden of malnutrition has been identified in developing countries such as in South Africa [21], Brazil [22], China [23], Vietnam [22], and Bangladesh and Indonesia [19]. At the population level, the double burden of malnutrition was documented among adolescent girls in seven African countries [24]. One study looked at Egyptian infants dealing with the double burden of malnutrition in the form of micronutrient deficiencies, stunting and overweight condition [25]. In Southeast Asian countries, children are experiencing the double burden of malnutrition where prevalence of stunting and underweight condition is still high, but prevalence of overweight condition and obesity is increasing at the same time [26,27,28].

As seen in Figure 1, most developing countries undergoing nutrition transition suffer from the double burden of malnutrition at varying prevalence. A recent study revealed that the double burden of malnutrition, as measured by child underweight and adult overweight condition, had already affected almost all 100 developing countries used in the analysis [29]. A review of 28 studies revealed that the problem exists all over the world, from the Asia Pacific region to Latin America and Africa [29]. In a Mexican population study, the prevalence of the double burden of malnutrition was 6.2% when measured as a coexistence of maternal central adiposity and child stunting (MCA) [30], and 8.4% when measured as SCOWT [15]. The prevalence of SCOWT was higher in other Latin American countries such as Argentina [31] with 12%, Ecuador [32] with 13.1%, and Colombia [33] with 13.2%. The prevalence of double burden of malnutrition was relatively low in several developing countries in Latin America, such as Brazil [34] and Chile [35]. In Asia, a rural population study documented the prevalence of maternal child double burden (MCBD) at 11% in Indonesia and 4% in Bangladesh [19]. In Malaysia, the prevalence of the double burden of malnutrition, measured as underweight child and overweight mother pairs, was 29.6% in the general population [36], and 12.5% among indigenous people of Peninsular Malaysia [34]. When measured as SCOWT, the prevalence of double burden of malnutrition in the indigenous Malaysian population was 19.4% [37]. These numbers clearly show that the double burden of malnutrition is an emerging problem in developing countries.

The increasing prevalence of double burden of malnutrition across studies in the same country highlights the emergence of this problem (Figure 2). Data from the Indonesia Nutrition Surveillance System showed the prevalence of the double burden of malnutrition in rural Indonesia was 11% in 2000–2003 [19]. In 2013, a study using 15-year panel data from 1993 to 2007 showed that the prevalence of double burden of malnutrition in Indonesia was 16% [38]. A study involving 12,048 households in Indonesia estimated the prevalence of the double burden of malnutrition at 19% [39]. A study in rural Indonesia documented the prevalence of double burden of malnutrition at a staggering 30.6%. This figure was understandably higher because the cut-off for maternal overweight used in the study was relatively low, at BMI > 23.5 [40]. Additionally, a study from an urban area revealed that the number of double-burden households was 24.7% with 36.4% of children experiencing stunting and 70.2% of mothers being overweight/obese [41].

In 2005, a Guatemalan study revealed that the prevalence among child and mother pairs consisting of stunted children and overweight mothers was 16% [15]. This number was slightly higher in a 2012 study, where the SCOWT prevalence was 16.8% [17]. A 2014 publication reported 20% of Guatemalan households as suffering from SCOM [18]. Two previous literature reviews have focused on the double burden of malnutrition. The first emphasized differences in the levels [13]. The second provided an update and additional analysis of the double burden of malnutrition as measured by underweight child and overweight/obese adult pairs using the WHO countries dataset [29].

Due to the apparent complexity of the double burden of malnutrition, we sought to describe and frame the problem in a logical, comprehensible manner that would enable policy makers to take necessary action. One approach widely used to describe such a complex phenomenon is the socio-ecological model (SEM). The socio-ecological approach to public health originated from the concept of the ecology of human development introduced by Bronfenbrenner in the 1970s [42]. He argued that along with human growth and environmental changes, multiple systems of interaction occur that are not limited to a single setting, but further involve environments beyond the immediate situation [42]. Given that most public health challenges are complex, the SEM has been widely embraced in the field of public health [41,42]. Numerous studies have applied Bronfenbrenner’s ecological model to understand complex health problems, such as sexually transmitted disease [43], mental health [44], and prevention of childhood obesity [45]. The SEM offers a more holistic approach that acknowledges interrelations among multiple levels of influence that impact health outcomes [46]. Levels described in the SEM are individual, interpersonal, community, organizational, and at the policy level.

Our literature review will offer a new insight into correlates of the double burden of malnutrition through the use of the socio-ecological model. This is a good starting point, because it identifies significant correlates and characteristics of the target population most in need of intervention. With its multiple levels and interrelationships, we believe the socio-ecological model is best for capturing the complexity of the double burden of malnutrition that is hampering populations in developing countries. The SEM’s underlying premises arguably are suitable for examining the phenomenon of double burden of malnutrition and interrelations among correlates, or determinants, on multiple levels. Hence, this review aims to summarize present knowledge about correlates of the double burden of malnutrition based on the socio-ecological model (SEM).

## 2. Materials and Methods

In this literature review, we did not limit ourselves to specific levels and measures of the double burden of malnutrition, but placed equal emphasis on correlational evidence of this phenomenon. Relevant studies were systematically identified using a structured search within the PubMed database. Initial keywords searched were “double burden of malnutrition,” “dual burden of malnutrition,” “stunted child or overweight mother,” and acronyms related to the double burden of malnutrition such as DBM, SCOWT, or SCOM. We did not impose date restrictions, but limited our search to articles published in English. Reference lists from identified articles were examined (hand searches) for additional relevant studies missed by electronic database indexing. In terms of the type of the study, we did impose restrictions for study inclusion criteria in the analysis: The minimum level of evidence was a cross-sectional study; for a study to be relevant and, therefore, included, we required reporting of risk factors or protective factors for the double burden of malnutrition. Articles were excluded if the study was not conducted in developing countries. We conducted a systematic review of all articles published between January 2000 and January 2016. As of 19 January 2016, the aforementioned keywords turned up 1806 articles; however, only 15 of those articles contained correlates of the double burden of malnutrition identified in this review. The records were screened at the abstract level. Then, in July 2019, we updated the literature review with additional 7 relevant articles published after January 2016. Figure 3 shows the flow diagram of studies included in the literature review. Two persons were involved in the search process independently and then compared with each other to determine whether a study should be involved in the analysis. In the absence of correlates on specific levels of the SEM that are associated with the double burden of malnutrition, we used correlates of overweight and obesity to complete the narrative.

## 3. Results and Discussion

### 3.1. Socio-Ecological Model of Double Burden of Malnutrition

As seen in Table A1, we extracted and analyzed correlates of the double burden of malnutrition in the context of the socio-ecological model initiated by Bronfenbrenner and later widely adapted for health promotion interventions [47]. Figure A1 illustrates the socio-ecological model pertaining to the double burden of malnutrition in developing countries. The individual level is at the model’s core, and forms a layer of personal attributes related to the double burden of malnutrition. Individual-level correlates include: Race and genetics; in-utero adaptation; maternal short stature; breastfeeding status; and low maternal education. The interpersonal level represents direct influences or interactions between individuals. Related to the double burden of malnutrition, interpersonal correlates are: Family size; household food security; and dietary diversity. At the organizational level, the workplace environment is a potential correlate of the double burden of malnutrition. At the community level, correlates of the double burden of malnutrition are both rural and urban areas. All of these levels are bound to policies and systems in place. For example, government and political structure, social structure, public policy, health care and food systems, economy, and media are all a part of larger policies and systems that influence individual behavior. Policies and systems associated with the double burden of malnutrition are: Economic development; food policy; and urbanization.

### 3.2. Individual Level

#### 3.2.1. Race and Genetics

Evidence shows that racial differences are associated with different rates of the double burden of malnutrition. Studies among Latin Americans revealed that being from the indigenous population was associated with higher risk. The prevalence of coexistence of maternal central adiposity and child stunting (MCA) in a Mexican population was higher among indigenous families [30]. A study in Guatemala showed that being from an indigenous population doubled the risk of SCOWT (odds ratio (OR) = 2.0, 95% CI = 1.3–3.1) [48]. This study was supported by another study from Guatemala that highlighted higher prevalence in SCOWT pairs in indigenous households (28.2%) than non-indigenous households (14.4%) [18]. The observed differences might be attributable to variations in body fat storage between indigenous or non-indigenous populations, race or ethnicity. Because of important body composition differences that can lead to shifts in BMI–disease patterns, the WHO has recommended reducing the BMI cut-off for overweight status and obesity in Asian populations [49]. Evidence shows that, compared to Western populations, Asians have a higher cardiovascular risk at any given level of BMI [50].

#### 3.2.2. In Utero Dysadaptation (Barker Hypothesis)

Maternal health status and lifestyle, including both under- and over-nutrition arguably play a key role in programming metabolic risk in the offspring. In the late 1980s, Barker postulated that maternal nutritional condition during pregnancy can “program” the fetus for the development of chronic disease later in life [51]. The fetal origins, or Barker, hypothesis stated that under-nutrition and an unfavorable intrauterine environment at critical periods in early life can cause permanent changes (in both structure and function) in the developing systems of the fetus [52]. These systemic changes may manifest as disease over a period of time due to “dysadaptation” with changed environmental circumstances [52].

The double burden of malnutrition may provide support for the Barker hypothesis. Embryonic and fetal growth starts with proliferation, organization, and differentiation of the embryonic cells, followed by continued growth and functional maturation of the different fetal organs and tissues [53]. This process depends on the genetic profile of the embryo, the maternal-placental-fetal unit, adequate nutrients and oxygen supply to the developing fetus, maternal prenatal weight, and maternal weight gain during pregnancy [53]. When undernourished, Barker postulated that the fetus will adapt to this adverse in utero condition by using limited nutrients more efficiently [51]. Even so, the outcome of the pregnancy is likely to be a malnourished child. Later in life, with changes in external conditions, such as an influx of energy-dense food readily available for the formerly stunted infant, catch-up growth takes the form of body weight, rather than height. Hence, in a Latin American population, adults were getting shorter, but BMI increased, resulting in overweight status and obesity [54]. This finding provides evidence for the hypothesis that adult disease originates through fetal growth and results in permanent changes to endocrine and metabolic processes [55].

#### 3.2.3. Breastfeeding Status

Breastfeeding behavior has been found to protect against the double burden of malnutrition. In Indonesia, for a child who was breastfed, the odds of having MCDB was lower (OR = 0.84) compared to a child who was not breastfed (95% CI = 0.81–0.87) [19]. The protective effect of breastfeeding was even greater in a Bangladeshi population, where the odds ratio (OR) of having MCDB was only 0.55 (95% CI = 0.52–0.58) [19]. During the first four to six months of age, an infant who is breastfed exclusively grows faster than those who are not breastfed. [56] Compared to formula-feeding infants, the breastfeeding of infants also protects against childhood obesity by providing a higher level of leptin, which results in better control of the satiety signal, food intake and adiposity [56]. Breastfeeding for a minimum period of 30 days had a protective effect against overweight in preschool children of the semiarid region of Alagoas, Brazil [57]. Infant and young child-feeding (IYCF) practices are also a preventive strategy to stunting [58]. Breastfeeding practice is influenced by maternal knowledge and attitude, among other factors. Mothers with good literacy, positive attitudes toward breastfeeding, and an encouraging social environment will have better breastfeeding practices, which will prevent stunting in the near future [59].

#### 3.2.4. Maternal Short Stature

Ample evidence shows that short maternal stature is associated with child stunting, but evidence associating it with double burden of malnutrition was limited. A study in Brazil revealed that children of mothers who have a height less than 145cm had lower height-for-age Z-score (HAZ) than children of mothers more than 160 cm tall (*p* < 0.0001) [60]. A study in Uruguay showed that maternal short stature of <160 cm was a significant predictor for child stunting [60]. This result aligns with a previous study in Brazil that highlighted the association between short maternal stature with low birth weight (<3000 g; *p* = 0.01) and stunting (*p* = 0.019) [57]. In another population, Mexican children with a mother below 150 cm tall were 3.6 times more likely to be stunted than children with a mother who is taller than 150 cm [61]. Although the author mentioned that the settings being studied had experienced the double burden of malnutrition, the association of maternal short stature was only analyzed in relation to child stunting, and not maternal nutritional status; hence, the association with double burden of malnutrition in the previously mentioned studies [57,60,61,62] was hypothetical. A study using a large sample in rural Indonesia and Bangladesh revealed an association between maternal short stature and maternal child double burden (MCBD) [19]. The Indonesian data showed households that have a mother of short stature showed an increased odd of MCBD 2.32 times (95% CI = 2.25–2.40) higher in comparison to those households without short maternal stature. Similarly, data collected in a Bangladeshi population showed a parallel OR = 2.11 (95% CI = 1.96–2.26) for short maternal stature [19]. This evidence aligns with the results from a study in Guatemala that revealed maternal short stature as a risk factor for SCOWT (OR = 3.1, 95% CI = 2.1–4.7) [48].

#### 3.2.5. Low Maternal Education

Similar to other correlates of double burden of malnutrition on the interpersonal level, maternal education was extensively studied as a predictor of child stunting, but was rarely studied as a predictor of double burden of malnutrition. Low maternal education has been linked with child stunting [62]. Furthermore, a lack of maternal knowledge and certain attitudinal factors could eventually affect children through feeding practices [63]. The mother’s motivation, behavior, and self-efficacy have an important role in fulfilling school-aged children’s nutrition [63,64,65]. A maternal education intervention of 12 months’ duration toward the specific task of improving children’s intake, such as complementary and responsive feeding education, significantly reduced child stunting (OR = 0.19, 95% CI = 0.0–0.04) among rural Indian toddlers [66]. A quasi-experimental community-based trial providing mothers with biweekly group activity accompanied by monthly home visits significantly reduced the risk of child overweight condition (BMI-for age >85th percentile) but did not significantly reduce child stunting [67]. A cross-sectional study in 685 children under 5 years of age and their mothers in a poor urban area of Indonesia found that household with SCOWT pairs had lower nutrition literacy [68]. Related to the double burden of malnutrition, analysis of the Demographic Health Survey (DHS) dataset from 18 lower- and middle-income countries showed that a low level of maternal education increased the likelihood of SCOWT [69]. A study among an indigenous population in Malaysia showed contradictory results to those discussed above, whereby having mothers with high education increased the likelihood of SCOWT (OR = 1.7, *p* < 0.05) [37]. Although more research is warranted regarding the importance of maternal education as a correlate of the double burden of malnutrition, the direction of its relationship appears to align with how it may impact child stunting.

### 3.3. Interpersonal Level

#### 3.3.1. Family Size

One correlate of the double burden of malnutrition identified in the interpersonal level was family size. A study in an Argentinian population revealed that households with SCOWT tended to have more people living in the house [31]. Large family size also correlated with MCBD, where having more than four people in the family increased the odds in both Indonesian (OR = 1.34; 95% CI = 1.28–1.40) and Bangladeshi populations (OR = 1.94; 95% CI = 1.77–2.12) [19]. This large family size was eventually the result of higher maternal parity and high number of siblings in the household. A study in Guatemala showed that households where the mothers have a higher parity have higher likelihood of experiencing SCOM (OR = 1.2, 95% CI = 1.1–1.3) [48]. More siblings in the household were also reported to be a significant predictor of SCOWT in a study involving lower- and middle-income countries [69].

#### 3.3.2. Household Food Security

While food insecurity was largely associated with child stunting [9,17], evidence showing food insecurity as a predictor of double burden of malnutrition remains scarce. Indirectly, ecological evidence showed that food insecurity was not necessarily associated with double burden of malnutrition. A study in 2012 revealed that the prevalence of double burden of malnutrition was the highest among the middle quintile of per capita consumption (22.7%) despite the fact that the prevalence of child stunting was the highest among the first quintile (70.4%), and maternal overweight status was the highest among the fifth quintile (53.2%) [17]. Arguably, in the third quintile of per capita consumption, there was not enough of an indication that the households experienced a form of food insecurity. One study found that household with SCOWT had higher food insecurity compared to normal. Moreover, the study revealed that although SCOWT household had sufficient ‘food quantity’ it was lacking in ‘food quality’ [68]. Furthermore, we hypothesize that in contrast to child stunting, double burden of malnutrition is likely to occur in households that possess some form of food security or have access to an adequate quantity of food, but food may lack nutrient quality.

#### 3.3.3. Household Dietary Diversity

At the interpersonal level, a correlate of the double burden of malnutrition was partly represented by household dietary diversity. Some studies have shown that dietary diversity score (DDS) is positively related to weight-for-age Z-score (*p* < 0.001), height-for-age Z-score (*p* < 0.005) [70] and that the higher the dietary diversity, the lower the likelihood of double burden of malnutrition [71]. A study in rural Malaysia revealed that dietary diversity in households with an underweight child and overweight mother pair was relatively low [36]. It is also supported with another study in Indonesia which found that children who lived in a poor dietary diversity household had 5 times greater risk of being stunted [72]. In contrast, households with higher dietary diversity scores were associated with lower likelihood of child stunting [73].

Among the diverse food groups, foods containing high growth-promoting nutrients are essential to prevention of the double burden of malnutrition. A cross-sectional study in an Indonesian population showed that when consumption of the “high-animal products” was in the highest quartile, the risk of maternal-child double burden decreased (aOR = 0.46; 95% CI = 0.21–1.04) relative to those in the lowest quartile [40]. The high-animal products are specifically high in growth-promoting nutrients such as protein, calcium and zinc.

### 3.4. Organizational Level

#### A Shift in the Workplace Environment

There is a dearth of evidence associating double burden of malnutrition with a shift in the workplace environment, with limited opportunity for physical activity. Changes in the environmental settings and type of work were reported to be a significant factor in the increasing rate of overweight condition and obesity [74]. The amount of physical activity in the workplace declined with the shift from physical labor to sedentary work [74]. Work-related physical activity and BMI among an Indonesian population showed that those with sedentary work were the highest percentage of overweight people, even compared to those who were unemployed [38]. Furthermore, Roemling and Qaim showed that the prevalence of obesity was similar between people with light work, sedentary, and housekeeping work [38]. This illuminates the notion that a housewife who does mostly housekeeping work is at risk of overweight and obesity. If the effort to eliminate child malnutrition were failing in a country, inactivity among housewives might be a driving force of the increasing prevalence of the double burden of malnutrition [16].

### 3.5. Community Level

#### Rural and Urban Area

In the socio-ecological model, both rural [17] and urban settings [17] are associated with the double burden of malnutrition. Prevalence, measured as SCOM, in rural Guatemala in 2000 was 19%, while in the urban area was 13.4% [17]. The adjusted OR showed no significant difference in this population [17]. In urban areas, the increased prevalence of overweight condition and obesity drive the increased rate of double burden of malnutrition. The incidence of overweight condition and obesity was higher in urban than rural areas in Indonesia [27,38], Mauritius [75], Malaysia [76] and Cambodia [77]. However, the increase was not exclusive to urban areas. Increased prevalence of overweight condition was observed in both poor rural and urban women in Bangladesh [78].

Contrary to the popular belief regarding an “obesogenic” environment being largely embedded within urban settings, evidence from Latin American countries suggests that double burden of malnutrition was more prevalent in rural settings. A study about the double burden of malnutrition in Columbia showed that the prevalence of child stunting was highest in rural areas [79]. Similarly, the prevalence of coexistent maternal central adiposity and child stunting (MCA) in a Mexican population was higher in rural areas [30]. These studies might be biased by large indigenous populations in rural areas, which are prone to experiencing the double burden of malnutrition.

### 3.6. Policy

#### 3.6.1. Economic Development

The double burden of malnutrition is believed to affect populations undergoing economic improvement, as opposed to low-income populations. The association between the double burden of malnutrition measured as SCOWT and per capita GDP supports the hypothesis that economic development increased the likelihood of SCOWT [80]. Analysis of the DHS data from Latin American countries showed that the prevalence of stunting was the highest in poorer countries, while the prevalence of overweight condition was highest in countries with growing economies [13]. There is growing evidence that the problem of double burden of malnutrition in a household affects mostly countries with a middle level of gross domestic product (GDP). The prevalence of double burden households is highest in the middle gross national product (GNP) countries [22]. The double burden of malnutrition arguably started among rich populations, but has shifted toward the poorest population group in Indonesia [38]. At the household level, a study in Indonesia showed that the double burden of malnutrition was more prevalent in households with a high socioeconomic status (SES) [39]. A study of indigenous populations showed opposite results [17,36]. A study in indigenous Peninsular Malaysia showed that households with income per capita of less than USD 29.01 were associated with increased risk of SCOWT [37]. A similar finding was observed among indigenous Guatemalans, where SCOWT pairs were more prevalent among the low and middle SES households [17]. Furthermore, multivariate logistic regression models showed that households at the middle consumption quintile have a higher likelihood of having SCOM, compared to households in the first quintile (OR = 1.74; 95% CI = 1.13–2.67) [17].

#### 3.6.2. Food Policy

To the best of our knowledge, no studies have specifically addressed food policy as correlates of the double burden of malnutrition. With the open-market policy and globalization, some argue that the power of transnational industries will affect the pace of nutrition transition in countries with flourishing economic development. Furthermore, as countries undergo the nutrition transition, the problem of double burden of malnutrition is likely to increase. The increases in obesity for the last 3 to 4 decades in almost all countries seem to be driven mainly by changes in food composition and availability [81] through the global food system that produces more processed and affordable food [4]. Data from the FAO in 2006, collected from six developing countries, showed that there was an increasing trend in dietary energy availability from 1970 to 2000 [12].

The problem with an influx of relatively affordable energy-dense foods has been negated in several countries through food taxation policy. Edible oil-pricing policies were changed a number of times in China between 1991 and 2000 to influence dietary composition toward less fat [82]. Other food taxation such as the soda taxation policy in Mexico since 2014 showed that a 10% increase in the price of a soda beverage decreased consumption by 10–13% [15]. Such policies provide promising tools to prevent obesogenic environments, but challenges from the food industry and implications for international trade are significant. Since the problem of the double burden of malnutrition is partly made up from the rise of overweight and obesity, food policies in developing countries should also emphasize as potential strategies inhibiting obesogenic environments.

#### 3.6.3. Urbanization

Popkin has argued that the double burden of malnutrition in one household is related to urbanization [83]. As urbanization happened, households increased their level of income and food became more available in greater quantity, but not in quality. In China, urbanization was solely responsible for decreased daily energy expenditure by about 300 to 400 kcal per day, while going to work by car/bus contributed a 200 kcal per day reduction in calorie requirement [84]. The characteristics of an “urban” diet were calorie-dense foods, dominated by fat and sugar [85]. A study in urban poor settings in Kenya indicated that one of the distinct characteristics of urbanization was a high reliance on street food that was energy-dense [86]. The food was high in energy but low in micronutrients and protein, affecting child growth, especially height. When children did not receive enough micronutrients and protein in the first two years of life, they had a tendency to become stunted. Mothers who consumed high-energy food accompanied by low physical activity tended to be overweight.

### 3.7. Potential Intervention Strategies

Using the socio-ecological model, we suggest several targets and settings for an intervention aimed at the double burden of malnutrition (Table A2). First, we emphasize that tackling the problem of the double burden of malnutrition should be seen as a holistic process. Multiple levels of the SEM approach should be addressed at the same time, because of the inter-connectedness among levels. Developing the interventions should include consideration of the socio-cultural context, including cultural practices, social support, political and physical environment, and also individuals’ motivation and capabilities to change [87].

At the individual level, we argue that the mother plays a key role in preventing the double burden of malnutrition for herself and her family. Hence, empowering women to take care of the household in the direction that supports healthy food choice and availability is seen as critical [38,77]. Evidence shows that maternal education mitigates the rate of the double burden of malnutrition [88]. Therefore, educating women will lay the groundwork for follow-up interventions targeting women who may become mothers. Improvement in nutrition literacy is among the recommended strategies for addressing the double burden of malnutrition among the Malaysian population [37]. An educated mother will be more receptive to programs and interventions such as the 1000 days of early life, safe motherhood, and balanced diet. When mothers understand the benefit of proper nutrition during the 1000 days of early life, they may be more mindful of providing nutritious food for themselves starting from pregnancy through the lactation period [89]. Attempts should be made to break the cycle of malnutrition, seen in conditions such as child stunting.

Providing intensive 12-week nutrition education in obese/overweight mothers with stunted children was successful for engaging mothers in physical activity and also improving children’s fruit, vegetable and animal protein consumption in Indonesia [78,79,90]. Evidence has shown that maternal short stature was associated with increased likelihood of the double burden of malnutrition [19,48]. Hence, improving nutritional awareness, knowledge, self-efficacy and outcome expectation may play a key role in behaviorally oriented nutrition education [76,77], and along with food fortification and supplementation [91] strategies will benefit the health of the future generation.

At the interpersonal level, inequalities in food distribution within the household should be minimized and healthier food for all should be promoted. A study in Indonesia revealed that residing in an urban location increased intra-household nutritional inequality [38]. Similar results were shown in an urban population in Mysore, India, where the energy intake was more or less equal among family members, but parents tend to consume more adequate protein compared to children [92]. In addition to food distribution, emphasis should be placed on improving the type and quality of foods brought to the family table. Household dietary diversity should be promoted by emphasizing differences in nutritional needs of each family member. For example, to prevent the double burden of malnutrition, children should be given foods high in the growth-promoting nutrients such as animal products to prevent stunting [40], while mothers should be encouraged to eat healthy food such as fruits and vegetables to prevent overweight and obesity [93]. Family planning programs should be encouraged and promoted because evidence shows that a large family is associated with double burden of malnutrition [46,61].

At the organizational level, modernization has led the workplace to be more sedentary than the previous decades [74]. Some study authors have been proposing strategies for minimizing sedentary time during working hours such as intermittent physical activity breaks by walking to get a drink of water or a snack [82,83]. In a recent study, excessive time spent in front of the TV, computers and video games (known as screen time) was significantly associated with BMI gain [94]. Therefore, in developing countries where most mothers are housewives, enormous amounts of screen and sitting time should be reduced. A study in Malaysia showed that being a housewife was associated with physical inactivity with adjusted odds ratio (OR) of 1.78 (95% CI = 1.56–2.03) [95]. Having a peer group committed to doing physical activity in their spare time could encourage mothers to focus on their health.

At the community level, changes in the built environment are inevitable due to economic development and mechanization in both rural and urban areas. The double burden of malnutrition was reported in both of these areas, partly due to the increase of obesogenic environment [17]. In rural areas, advances in technology and mechanization have reduced the amount of physical labor required for many jobs and have decreased the activity levels of many workers [84]. To account for the decline in caloric expenditure derived from physical labor, physical activity should be promoted as part of a healthy life [96]. In urban areas, an increase in the availability of convenience foods is a contributing factor in the rise of overweight and obesity [97]. High rates of urbanization and the growth of large cities may be a double-edged sword, because it allows for the expansion of retail marketing of healthy and unhealthy foods [88]. Unfortunately, many convenience foods, either from street vendors or fancy food chains are high in calories [86]. Consumption of high-calorie foods combined with more sedentary work common in urban areas has increased the prevalence of overweight condition and obesity in developing countries.

The indigenous community requires special consideration because of the peculiar pattern of correlates of double burden of malnutrition. Interestingly, indigenous populations in rural Latin America such as Mexico [30] and Guatemala [17] are at higher risk for the double burden of malnutrition. Similarly, in an indigenous population in Peninsular Malaysia higher risk of the double burden of malnutrition was associated with lower per capita income [37]. Whether this anomaly was due to genetic traits in the indigenous population has yet to be examined. Nonetheless, providing culturally accepted nutrition education to indigenous populations living in rural areas might be a good starting point for an intervention.

At the policy level, several countries have implemented a taxation policy for energy-dense foods such as vegetable oil and soft drinks. This policy should be maintained and expanded to other unhealthy foods to reinforce community-level interventions to address healthy eating. For example, the Chinese policy to increase the price of edible oils might trigger a shift in dietary composition from fat towards proteins and complex carbohydrates [82].

To prevent a further influx of urbanization, rural areas should be developed equally with, if not more than, urban cities. Countries undergoing urbanization in Africa, such as Kenya [97] and Egypt [25] should promote dietary diversity and quality while implementing policies to address excessive consumption of energy-dense junk foods. One strategy proposed to prevent the double burden of malnutrition was the reduction of inequality and poverty [88]. More well-paid jobs in rural areas would minimize inequality and slow urbanization.

Economic development experienced in some middle-income countries provides a good opportunity for food security, but food security assessed as the availability of food in quantity alone might not be adequate to prevent the double burden of malnutrition. Unlike households facing only the problem of child stunting, households undergoing the double burden of malnutrition seem to have the capacity to access food, but have a problem choosing healthy foods that are lacking within the household. Food security in these households might promote the double burden of malnutrition when a household is spending money on less nutritious food. Consumption of energy-dense foods lacking in growth-promoting nutrients does not prevent child stunting but instead may lead to excess weight gain, overweight status and obesity. Therefore, providing mothers in charge of household food spending with the knowledge, skills-set, motivation, and self-efficacy of healthful eating can be influential in preventing the double burden of malnutrition. An intervention study targeting low-income, overweight and obese mothers to improve food choices, fat-consumption habits, and physical activity in their 1- to 3-year-old children, has shown promise [98].

In this literature review we aimed to describe correlates of the double burden of malnutrition using the socio-ecological model. Correlates are factors associated with the double burden of malnutrition, but not necessarily determinants that imply causation [99]. The majority of evidence relating to the double burden of malnutrition is brimming with findings of significant cross-sectional associations between a range of individual, interpersonal, organizational, community and policy-level variables. Since these were correlational studies, the observed relationships do not support causal inferences, but may suggest hypotheses for further study.

Because the double burden of malnutrition is a complex phenomenon, the socio-ecological model was successful in depicting the relationships among the correlates of the double burden of malnutrition across multiple levels. Currently, the evidence regarding correlates of the double burden of malnutrition is dominated by research at the individual and interpersonal levels. Much of the evidence for organizational, community, and policy levels is based on studies pertaining to overweight and obesity. Hence, some have argued that statistically, double burden of malnutrition such as SCOWT pairs are not independent from the components of maternal overweight status and childhood stunting but depend heavily on the prevalence of overweight mothers [16]. Others have argued that the SCOWT phenomenon can be best explained as a consequence of rapid secular increases in maternal weight [69]. Further study is warranted to illuminate whether the broader layers in the socio-ecological model such as rural and urban settings, a shift in the workplace environment, economic development, food policy and urbanization are indeed significant predictors of the double burden of malnutrition and not merely overweight status and obesity.

Even though some have suggested that the double burden of malnutrition in the form of stunted child and overweight mother pairs (SCOWT) is merely a statistical artifact [16], the health and economic consequences are real and devastating. The consequences of the double burden of malnutrition raise public health concerns, starting with the increase of the burden of disease due to under-nutrition. Simultaneously, over-nutrition will lead to an increase of non-communicable diseases (NCD) including obesity, cardiovascular disease, and hypertension. In fact, NCDs account for 80% of the total burden of disease mortality in developing countries; it is estimated that economic production of $84 billion in USD will be lost from heart disease, stroke, and diabetes alone [100]. Hence, it is assumed that the double burden of malnutrition is burdening the already inadequate and overextended health budgets of developing countries [101].

This narrative review is intended to describe correlates of the double burden of malnutrition in the context of the socio-ecological model. Once there is sufficient evidence to describe correlates on all levels of the SEM through a systematic review or meta-analysis, a more objective picture of the double burden of malnutrition will emerge. Limitations for this narrative review include the restriction of article inclusion in the English language and dependence on the coverage of the PubMed database. Some relevant studies and grey literature might not be captured using this search strategy, and that could affect the current narrative. Although evidence was limited for the double burden of malnutrition at the organizational, community, and policy levels of the SEM, this review explained the impact of those levels adequately using correlates of overweight status and obesity. Considering that overweight status and obesity are part of the puzzle in the double burden of malnutrition, we do not expect the effect of the correlates to be very different. We realize that by understanding the double burden of malnutrition through the socio-ecological model, our efforts to tackle the double burden of malnutrition cannot be spatial. Furthermore, we offer some potential intervention strategies based on the SEM that might be useful for future interventions.

## 4. Conclusions

The double burden of malnutrition is a public health phenomenon associated with an array of socio-ecological determinants. Implementation of an integrative model such as the SEM offers a potential solution for combating the complex problem of the double burden of malnutrition. Correlates at each level should be addressed holistically and efforts can target individuals, households, organizations, communities, and governments. Addressing malnutrition in all its forms requires an integrated approach to address the root causes of malnutrition at all stages of human development.

## Figures and Tables

**Figure 1 ijerph-16-03730-f001:**
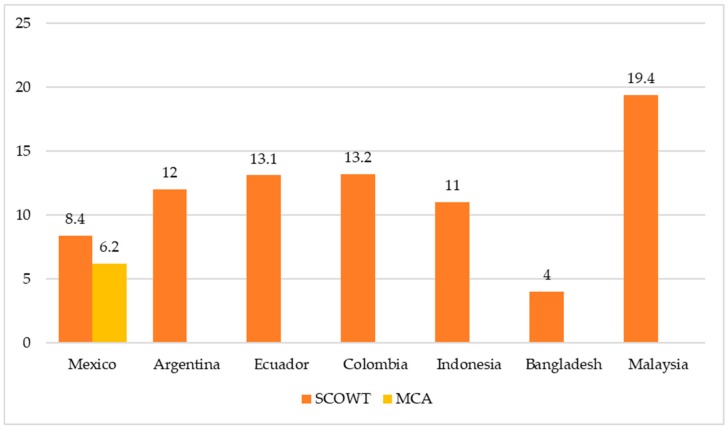
Prevalence of double burden of malnutrition (DBM) in developing countries.

**Figure 2 ijerph-16-03730-f002:**
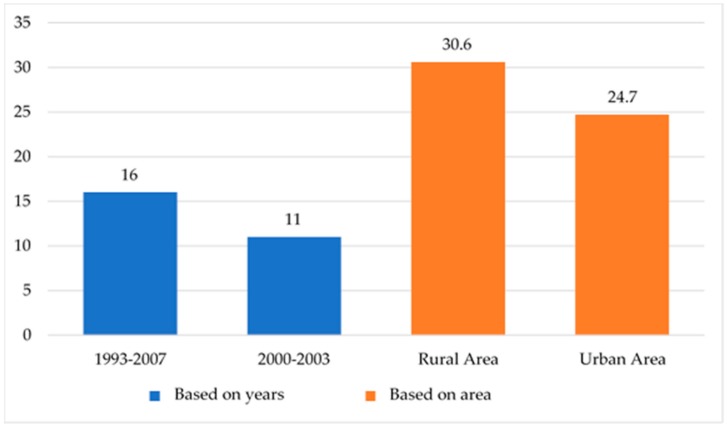
Indonesian DBM prevalence through the years.

**Figure 3 ijerph-16-03730-f003:**
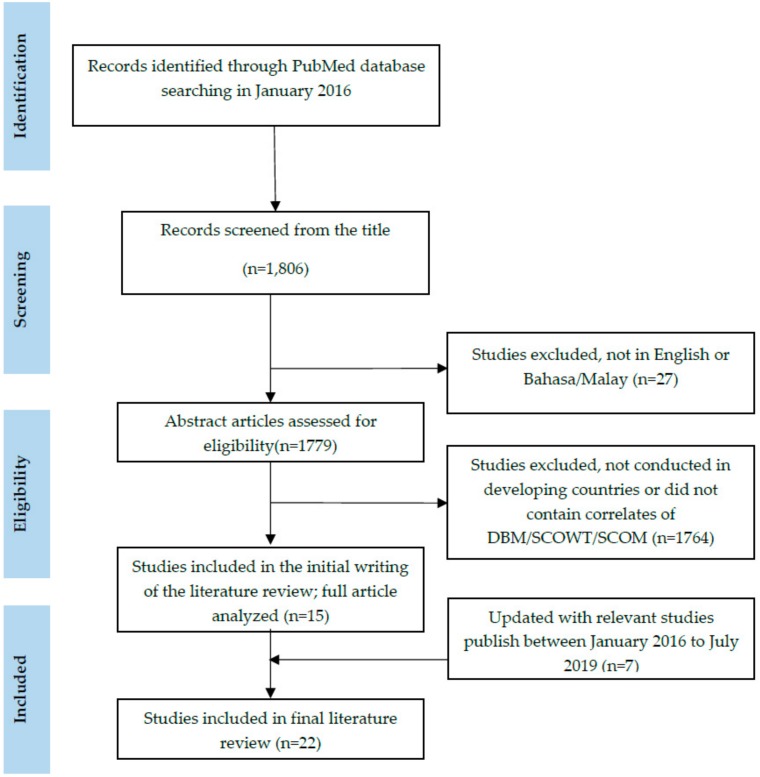
Flow chart of the study included in the literature review.

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
