# Peer review of "Socio-Ecological Model of Correlates of Double Burden of Malnutrition in Developing Countries: A Narrative Review"

_ijerph, 2019, doi:10.3390/ijerph16193730_

Round 1

Reviewer 1 Report

This manuscript focuses on the issue of double burden of malnutrition, which involves the coexistence of under- and over- nutrition within the same individual, household or population. Guided by the socio-ecological model (SEM), the authors reviewed the correlates of double burden of malnutrition. In the comments that follow, I offer some suggestions for improvement of the manuscript and some lingering questions.

Abstract

Line 13: should be “through” not “though.”

Line 16: the statement “were narrated critically” is a bit confusing. Consider rephrasing it.

Introduction

Like 35: the authors state “Among developed countries, USA had the highest BMI [4].” This statement would read better if the authors used the term “prevalence rate of obesity” rather than BMI.

Lines 36: the authors mention that the trend in obesity remains high and significant. It would be helpful if actual data was also provided here.

Line 43: “developing countries in Africa and Southeast Asia were relatively lower in overweigh and obesity…” Please rephrase to talk in terms of prevalence rates.

Line 60: “At the individual level, the problems of under- and over-nutrition 60 coexist in the same individual.” Please change to “may coexist.” The authors should try to avoid using absolute statements.

Authors do a good job in the Introduction discussing the prevalence of the double burden of malnutrition, but they do not at all discuss the SEM. Even though it is discussed further in the paper, since this approach is relevant to your analysis, please include at least a brief discussion on how it applies to your research question. Additionally, the last paragraph of the Introduction should summarize the goals of your research and include a few sentences about your contribution to the literature.

Methods

What was the time frame you focused on when conducting literature review?

Results

Line, 260: “Even though more study is warranted.” Consider changing study to research.

Other comments: Please proofread. There are many small grammatical and syntax errors that I did not point out, but they should be addressed by the authors. Also, please make sure that you are consistent with the tense that you use.

Reviewer 2 Report

This is a very interesting review paper discussing a great public health concern, double burden of malnutrition. The authors have used the widely adapted socio-economical model to capture relevant correlates related to the phenomenon. The topic is very relevant to the aim and scope of the IJERPH. Overall, the manuscript is well written and the rationale of conducting this narrative review is well-established. The use of Socio-economical model to go through the articles related to double burden of malnutrition is a novel idea and enhances an understanding of this complicated phenomenon. The Methods section, however, requires more specification and clarity, and the Results section is currently quite confusing to read as it does not specifically address the results of the study but includes much Discussion. I have provided my comments below:

INTRODUCTION

Introduction is well-written. However, I would encourage the authors to consider whether the section (lines 85 to 111) describing the prevalence of double burden of malnutrition could be described in a graph rather than in a text as it is quite heavy for a reader in its current form.

There are parts in the results section that I would rather see in the introduction, such as the description of the Bronfenbrenner’s model (lines 150 to 167).

METHODS

Was there any restrictions to the studies related to the type of study?

How were the records screened- on title level or on abstract level?

How many persons were involved in the search process?

A flow chart of the study would be informative.

RESULTS

As stated earlier, the results section in its current form resembles more like a discussion section as it includes many references that would be more suitable to the introduction or discussion. F example Barker hypothesis is not a result of your study, although it is very relevant to the topic. I would encourage the authors to go through the results section and include only the references listed in the table A1 (which by the way needs a title) in the results and move the rest to the discussion (or introduction where relevant) or either name the Results section as “Results and Discussion” (which would however be a bit confusing to the readers).

Overall, the contents of the manuscript are very relevant and well-written. The manuscript would however be much improved if the manuscript form would be corrected to either a traditional review paper with no methods described or a literature review following the format of introduction-methods-results-discussion-conclusions.    

Round 2

Reviewer 1 Report

Page 4, starting with line 160: “Our initial time frame when we did the literature review was between the year January 2000 to January 2016.” Change to “We conducted a systematic review of all articles published between January 2000 and January 2016.”

All the others comments have been adequately addressed by the authors. 

Reviewer 2 Report

The authors have addressed the comments previously made in enough detail. The manuscript has been improved and now warrants publication in IJERPH.